# Vaccine Hesitancy and Trust in the Scientific Community in Italy: Comparative Analysis from Two Recent Surveys

**DOI:** 10.3390/vaccines9101206

**Published:** 2021-10-19

**Authors:** Chiara Cadeddu, Martina Sapienza, Carolina Castagna, Luca Regazzi, Andrea Paladini, Walter Ricciardi, Aldo Rosano

**Affiliations:** 1Department of Life Sciences and Public Health, Università Cattolica del Sacro Cuore, Largo Francesco Vito 1, 00168 Rome, Italy; chiara.cadeddu@unicatt.it (C.C.); carolina.castagna01@icatt.it (C.C.); luca.regazzi01@icatt.it (L.R.); andrea.paladini03@icatt.it (A.P.); walter.ricciardi@unicatt.it (W.R.); 2Unit of Statistics, National Institute for Public Policies Analysis (INAPP), Corso d‘Italia 33, 00198 Rome, Italy; rosano244@gmail.com

**Keywords:** vaccine hesitancy, European Social Survey, scientific community, trust, vaccination, public opinion

## Abstract

Vaccination rates in Italy fell until 2015 because of unfounded safety concerns. Public education and a 2017 law on mandatory vaccination have boosted rates since then. The aim of our study is to explore how trust in the scientific community and attitudes towards vaccines have changed in the period of 2017–2019 in Italy. Data were extracted from the Italian section of the 2017 and 2019 editions of the European Social Survey (ESS). We compared the two surveys highlighting changes in public opinion on vaccines. A descriptive analysis of the socio-cultural variables according to the answers provided to key questions on the harmfulness of vaccines was conducted. Differences between percentages were tested by using the χ^2^ test. The association between the opinion about the harmfulness of vaccines and trust in the scientific community was analyzed through a logistic regression model. Compared to ESS8, ESS9 showed an increase in the percentage of respondents disagreeing with the harmfulness of vaccines. Trust in the scientific community raised in the period from 2017 to 2019 (59% vs. 69.6%). Higher education was significantly associated with disagreement regarding the harmfulness of vaccines (odds ratio (OR) = 2.41; 95% confidence interval (95%CI) 1.75–3.31), the strongest predictor was trust in the scientific community (OR = 10.47; 95% CI 7.55–14.52). In Italy, trust in the scientific community and in vaccinations has grown significantly in recent years, indicating a paradigm shift in public opinion compared to the past. Central actions and effective public communication strategies might reduce vaccine hesitancy and could be essential to garner public trust.

## 1. Introduction

Vaccines are widely recognised by health authorities and the medical community as a major tool for achieving public health benefits in terms of the prevention of infectious diseases, with a typical example being the eradication of smallpox in 1980 [1,2]. However, for many individuals, the demonstrated benefits of vaccines are not sufficient to embrace vaccination whole-heartedly. These individuals doubt the value of vaccines, worry over their safety, and question the actual need for them, an attitude called vaccine hesitancy. With this term, we refer to the “delay in acceptance or refusal of vaccines despite availability of vaccination services” [3], and it differs from what we call “vaccine refusal”, given that even those who get willingly vaccinated can harbour hesitancy towards certain aspects of vaccination [4].

Vaccine hesitancy is increasingly being considered as a health care priority because it threatens herd immunity and puts the lives of the most vulnerable (such as elderly or immuno-depressed patients) at risk [5]. Not surprisingly, in its recent recommendations to strengthen European Union cooperation on vaccine-preventable diseases, the European Commission has identified it as one of the main drivers of the huge drop in vaccination coverage [6].

Vaccine hesitancy is a complex phenomenon, context specific, varies across time, place, and types of vaccines [3].

Evidently, the most commonly cited reason for general population hesitancy towards vaccination is safety concerns. Furthermore, a lack of awareness and a low perception of the severity of preventable illnesses, traits that are also commonly reported among adolescents [7], and a strong belief in alternative medicine have often been cited as elements associated with hesitancy [4].

In previous research, we showed that when the Italian population is stratified on the basis of their attitude towards vaccination, those with anti-vax attitudes and who are skeptical towards the scientific community are mainly older males with low participation in political and cultural life and who are politically oriented to the right, whereas individuals who identify as being pro-vax/pro-science are represented by females with a certain level of involvement in political and cultural life and who are politically oriented to the left [8].

In addition, other recent international research has confirmed that beliefs, ideology, political orientation, and cultural background have a strong and statistically significant impact on vaccination-related behaviours [9,10,11].

As for ways to reduce vaccine hesitancy, it has been shown that science-supporting messages from experts about vaccine safety can lead to higher pro-vaccine evaluations. Particularly, when presented alone, the statistical information provided by an expert (i.e., the science-supporting message) about vaccine safety and effectiveness has produced stronger pro-vaccine beliefs than not presenting any science-supporting message (i.e., the control condition) [12].

From this point of view, Italy represents a good case study to understand how to properly address vaccine hesitancy. As a matter of fact, after rising for years, vaccination rates in Italy fell in the 2010s because of unfounded safety concerns, misinformation on the web and social networks, and anti-scientific judgments. The most dramatic decrease was the drop in the measles vaccine rate to 85%, which led to the re-ignition of the severe Italian measles outbreak in 2017, with 4991 measles cases (four of which were fatal) being reported [13].

As a response to this outbreak and in agreement with the 2010–2015 National Italian Immunization Plan, Italy’s Parliament gave the final approval to a law introducing a slate of mandatory vaccinations for preschool and school-aged children [14]. The implementation of the law boosted the interest of the media and of public opinion, contributing to an increased awareness of the importance of vaccination in the population. From this point of view, the communication and training activities for public health and healthcare providers implemented by local health units, national authorities, and scientific societies were also fundamental.

One of the broadest tools used to measure the attitudes, beliefs, and behaviour patterns, including questions on political preferences, religious practices, opinions on hot social topics (poverty, immigration, unemployment) in Europe is the European Social Survey (ESS). The ESS is an academically driven cross-national survey that has been conducted across Europe every two years since 2001 through face-to-face interviews with newly selected, cross-sectional samples that also investigates the socio-economic characteristics of respondents from 20 European countries [15].

In our previous research on this topic [8], we assessed opinions regarding the supposed harmfulness of vaccines in the Italian population by extracting data from the Italian section of the ESS conducted in 2017, with a particular interest in describing the interaction between opinions about the supposed harmfulness of vaccines and the perceived trust in the scientific community in regard to vaccines. The main of the present paper is to conduct a comparison of the results from the Italian section of ESS8 (data collected as part of the 8th round of the ESS) and ESS9 in order to explore how trust in the scientific community and attitudes towards vaccines have changed in the period of 2017–2019 in Italy. As a secondary aim of the study, we wanted to investigate the attitudes of the Italian people towards vaccination in a pre-COVID context.

## 2. Materials and Methods

Data were extracted from the Italian section of the last two editions of the European Social Survey (ESS), which was conducted in 20 countries in the years 2016–2017 (Round 8) and in 2018–19 (Round 9). In Italy, ESS8 was conducted between September 2017 and November 2017, while ESS9 was conducted between December 2018 and March 2019 on random samples of 2626 and 2745 people aged 15 and over, respectively. The surveys used the same questionnaires and methodology. More details about the surveys are available through the ESS website [15].

In particular, the analysis focused on certain issues concerning the social debate on vaccines, which was investigated with an “ad hoc module” that was used in both rounds of the Italian ESS, namely (1) ‘‘Vaccines are harmful and expose people to various diseases” and (2) ‘‘With regard to vaccines, recommendations from the scientific community can be trusted”. The level of agreement or disagreement with the statements was measured on a five-point Likert scale that included the options ‘‘Strongly agree; Agree; Neither agree nor disagree; Disagree; Strongly disagree”. The respondents could also refuse to answer or answer ‘‘Do not know”.

The characteristics selected to trace the socio-cultural profile of the interviewees were interest in politics; placement on political scale; opinion on the current state of health services in their home country; subjective general health; belief in any religion or cult; a description of the respondent’s domicile; participation in lectures, courses, or conferences; internet use and social trust; education; gender; and age.

### Statistical Analysis

A descriptive statistical analysis of the socio-cultural variables according to the answers provided to the key questions on the harmfulness of vaccines was conducted. Differences between percentages were tested by using the χ^2^ test.

The association between the opinion about the harmfulness of vaccines and trust in the scientific community, while adjusting for potential confounding factors (i.e., gender, age and education), was analyzed through a logistic regression model. For this purpose, the variable that identifies the opinion on the harmfulness of vaccines was dichotomized into respondents who agree or strongly agree and those who disagree or strongly disagree (used as a reference). Data from the two editions of the ESS surveys were simultaneously analyzed. A dummy variable identifying the ESS edition was included in order to evaluate whether time plays a relevant role in the model. We also included an interaction term between trust in the scientific community and time under the hypothesis that the effect of trusting the scientific community may significantly vary across the two editions of the survey in relationship with the outcome. Marginal effects of trust in the scientific community in the two ESS editions were also investigated.

## 3. Results

Out of the 2745 participants answering European Social Survey 9 (ESS9) in 2019, 53% were females and 47% males. The characteristics of the respondents were described as follows: 33% were between 45 and 65 years of age, 50% of them declared having between 0 and 12 years of education, 70% of them stated that they were not at all or hardly interested in politics, and 79% declared they believe in a particular religion. Moreover, the majority of interviewees came from country villages (43.4%), towns, or small cities (35%).

With regard to the harmfulness of vaccines, 63% of the respondents to the survey in 2018/2019 did not find vaccines harmful, whereas 14% thought that vaccines were harmful, and 23% were either undecided or did not answer. In comparison with the survey performed in 2017, ESS9 showed a slight decrease in the percentage of people agreeing that vaccines were harmful (19% in 2017 vs. 14% in 2019). On the other hand, the amount of participants disagreeing with the statement “vaccines are harmful” increased from 2017 to 2019 (50% vs. 63%) (Figure 1).

Concerning the attitudes towards the scientific community, the largest proportion of respondents declared that they do have trust in the scientific community (27% strongly agreed, 43% agreed), while 16% of them neither agreed nor disagreed, 6% disagreed, 2.5% strongly disagreed, 0.5% refused, and 5% did not know. Indeed, trust in the scientific community sharply increased from 2017 to 2019; in fact, among respondents in 2019, 69.6% agreed or strongly trusted the scientific community, whereas in 2017, only 59% trusted the scientific community (Figure 2).

In ESS9, most of the respondents who did not think vaccines were harmful had trust in the scientific community (88%) (Figure 3a), whereas 45% of the respondents who stated that vaccines were harmful had trust in the scientific community (Figure 3b). Comparing data from 2017 and 2019, an increase of +7% (*p* < 0.0001) can be noted in terms of people trusting the scientific community not declaring that vaccines were harmful.

Participants stating that vaccines were harmful mostly declared to have centre (48%) and right (34%) wing political orientations. Moreover, the respondents who believed that vaccines were harmful were characterized by attending less than 13 years of study (56% agreeing that vaccines are harmful vs. 46% disagreeing, *p* < 0.0001), never participated in congress, courses, or training (94% vs. 86%, *p* < 0.0001) and used the internet less often than their peers every day (46% vs. 54%, *p* < 0.0001) (Appendix A).

In respect to the 2017 survey, in 2019, even more participants who believed that vaccines were harmful rated their own health status as good or very good (70% vs. 61%, *p* < 0.001). On the other hand, giving a positive rating (scores 6–10) to the state of health service of the country was more frequent in participants disagreeing with the harmfulness of vaccines both in 2017 and in 2019, respectively, at 60% and 64% (Appendix A).

Comparing the data from 2017 to 2019, the proportion of males with no trust in the scientific community decreased (from 12% to 9%, *p* < 0.05). Similarly, an increase of the percentage of people having trust in the scientific community was observed in some subgroups of participants, including those living in big cities (65% in 2017 vs. 76% in 2019, *p* < 0.005) and those who were politically oriented to the right (62% in 2017 vs. 77% in 2019, *p* < 0.001) as well as in people with 5–14+ years of education (Appendix A).

Table 1 shows the results of a logistic regression that was conducted to evaluate the association between opinions about the harmfulness of vaccines and trust in the scientific community. Higher education was significantly associated with disagreement regarding the harmfulness of vaccines (odds ratio (OR) = 2.41; 95% confidence interval (95%CI) 1.75–3.31), but the strongest predictor was trust in the scientific community (OR = 10.47; 95% CI 7.55–14.52). The effect of this factor significantly changed between the two editions of the survey (OR of the interaction term = 1.73; 95%CI 1.10–2.83), with an increase in the marginal probability of disagreeing about the harmfulness of vaccines, both among those not trusting in the scientific community (from 33% to 34%) but, above all, among those trusting the scientific community (from 83% to 90%), as highlighted in Figure 4.

Additional Tables (Appendix A) are reported in Appendix A.

## 4. Discussion

The present work highlights some differences in the two rounds of the ESS surveys and shows how participant thoughts and attitudes changed over this time period, even if those changes were narrow. In 2019, there was a significant increase in the percentage of people who disagreed with the harmfulness of vaccines (from 50% to 63%). These results are also supported by the analysis concerning the Italian data conducted by de Figueiredo et al., which provided multiyear global-level estimates of vaccine confidence for 149 countries worldwide, exploring trends in confidence and the global determinants of uptake, including socioeconomic determinants and sources of trust. In these estimates, in November 2018, the percentage of Italian respondents strongly agreeing with safety and effectiveness of vaccines were 40−49.9% and 50−59.9%, respectively [16]. Furthermore, our findings indicated that the factors that were mostly associated with disagreement regarding the harmfulness of vaccines were higher education and, above all, trust in the scientific community. Several studies [16,17] corroborate the association between higher education and positive attitudes towards vaccinations, which is also confirmed by the results from the survey Special Eurobarometer 488, a cross-national longitudinal study designed to compare and gauge trends within the Member States of the European Union [11].

Indeed, trust in the scientific community was found to be quite high in most of the interviewees, resulting in increases in both the population that had positive attitudes toward vaccination and in those that asserted the harmfulness of vaccines.

Such increases might be interpreted in the light of the Italian socio-cultural context of the years 2017–2019. In June 2017, the introduction of the Decree Law (73/2017) resulted in higher vaccine coverage rates in childhood. This mandatory act probably increased the awareness of the population on the importance of vaccines and vaccination, highlighting the fundamental role of health scientists. In this context, the scientific community played a pivotal role in influencing public debate, especially regarding vaccinations, becoming a key player in the general political landscape [18].

Globally, the last 20 years have seen a major increase in the means of disseminating information, which are no longer linked to traditional tools such as posters and newspapers but that benefit from the use of the web and social media. Nowadays, the web is available and accessible in every part of the world [19].

Because of these changes, the scientific community has had to adapt to new media and particularly to the language of social media. In this context, many blogs, portals, and websites have been created in order to disseminate awareness, promoting healthy behaviours, including adherence to vaccination programs (e.g., Medical Facts, VaccinarSi) [20,21].

An example is the portal “ISSalute” [22], where the Italian National Institute of Health is conducting several initiatives in order to increase knowledge and confidence of the Italian general population towards vaccination. This portal translates the complex language of science, granting people useful and important information [23].

Trust from the public is a crucial issue for health care systems: it is correlated with patient satisfaction (which is arguably beneficial for health outcomes), and it helps to boost and maintain compliance with prevention policies [5].

Public health campaigns and news coverage aiming to promote vaccination should employ both scientific messages and pro-vaccine narratives that amplify the stories of individuals to better address the emotional aspect of misinformation [24]. Indeed, according to Daniele Chiffi (2021), mass health communication campaigns and the decision of how to frame the information in those campaigns are matters that are related to emotional and ethical issues: these external aspects are, as a matter of fact, capable of shaping the perception of risk and, as a consequence, of influencing the ultimate decision to be treated or not.

Particularly, emotion, in the sense of affect (a concept experienced as a state of feeling sensitive to the positive or negative quality of stimuli, i.e., linked to goodness or badness, happiness or sadness), has a major role in the perception and communication of risk. Affect appears to explain the inverse correlation of perceived risk and perceived benefit since if something such as vaccination is linked to positive affect, then vaccination-related risks are basically ignored.

On the other hand, if something is associated with negative affect, then risks are taken into account while benefits can be easily overlooked. Fundamentally, if an activity is presented by describing its rising levels of benefits (or risks), then the perceived level of risk (or benefit) decreases; on the contrary, if an activity is presented by describing its declining level of benefits (or risks), then the perceived level of risk (or benefit) increases. In brief, it appears that people make decisions not only based on what they think but also on what they feel [25].

Furthermore, in order to be successful, vaccination policy as well as other areas of public health requires strong collaboration across governmental and non-governmental institutions working together to achieve common goals [26].

For these reasons, to reach a wider audience, between 2018 and 2019, the Italian Ministry of Health produced two television spots featuring Ivan Zaytsev, a famous volleyball player, and Samantha Cristoforetti, an Italian astronaut. The spots highlighted the role that vaccination has played in their lives by protecting them from diseases, such as measles, and thereby supporting them in the pursuit of their professional careers in sports and space, respectively [27].

On the other hand and always referring to the scientific community, to better serve the increasing need to reach parents and healthcare workers with facts on immunization, in 2019, the Italian Society of Paediatrics and the Society of Neonatology set up direct telephone hotlines during the week. A neonatologist and a paediatrician were available to answer questions on any topic related to childhood vaccination, offering direct contact with citizens and resulted in being a very useful action to the increase trust towards vaccination and health in general [28].

Even though these initiatives were fundamental for volunteer citizen engagement, one of the most significant measures taken in response to the alarming reduction of vaccination coverage rates was the introduction of Decree Law n. 73/2017 (modified by the Law n. 119/2017). This law, which was implemented in 2017, extended mandatory vaccinations from four to ten for children up to sixteen years of age and for unaccompanied foreign minors.

An early evaluation of the impact of this law was promising: an Italian study highlighted how the mandatory act may have contributed to a prompt and substantial increase in vaccine coverage (VC) rates in childhood in 2019 compared to the year 2016, before the introduction of the law. Authors suggested that vaccine mandates for children may have resulted in public attention and discussion that, to some extent, also boosted the adherence to recommended vaccinations that are offered actively at the vaccination point of care during the immunization session. They also stated that, in addition to mandates, education programs encouraging responsible behavior and enhancing vaccine literacy should also be taken into account [29].

The ongoing COVID-19 pandemic has fostered scientific innovations in several medicine and public health areas and has provided new opportunities as well as new challenges for all the most hit countries to further strengthen their scientific culture. This has highlighted the importance of trusting scientific culture development and the scientific community as powerful tools to overcome epidemics and pandemics [30].

Since the rapid development of vaccines against COVID-19 is an extraordinary achievement and since successfully vaccinating the global population is the paramount challenge in order to control the pandemic, trusting the scientific community in this field has become vital [31].

In their article, Sturgis et al. suggest that, as with the protective effects of vaccines, public trust in immunization programmes is related to factors working at the community level as well as the individual level: the positive correlation between trust in science and vaccine confidence is stronger in countries with a high level of consensus regarding the reliability of science and scientists compared to countries where the level of social consensus is weaker.

Even if Sturgis et al. collected data before the start of the COVID-19 pandemic, they stated that the core findings could be evident in the specific case of coronavirus vaccination [17].

The results of the aforementioned study corroborate our findings that trusting the scientific community can have an important positive influence on attitudes towards vaccines. As such, it is imperative to focus further research on the identification of the factors that are involved in the creation of science-based societal consensus in order to steer effective public communication strategies about vaccination programmes.

However, this study should be considered in the light of some limitations, which are mostly similar to those highlighted in our previous paper [8]. Firstly, the key question of vaccine hesitancy has been formulated in a direct way and specifically for the ESS survey, so it is not based on a standardized questionnaire for the scope. Secondly, the aforementioned question is quite generic and does not refer to any specific type of vaccination (e.g., childhood vaccinations, influenza vaccination, HPV) and is therefore subject to personal interpretation by the respondents: for example, the parents of children may have only considered childhood immunization, while elderly could have replied after only considering influenza vaccinations, which may not render the same answer. In fact, should belief in harmfulness of vaccines depend on the perceived type of vaccine, a systematically different interpretation of the question according to age (or any other personal variable) would influence the results. It is therefore imperative that future studies in the field specify which vaccinations are considered when constructing questions. Thirdly, the scope of our study is limited by the restriction to only consider information included in the ESS survey. Even if the ESS survey collects information about a broad range of attitudes, beliefs, and behaviours in various socio-cultural groups, the fact that additional information could have been useful in describing the profile of Italians regarding their beliefs towards vaccination cannot be ruled out. However, strong evidence has been reported that the association between the factors investigated here (the role of the scientific community and increased vaccine confidence) may be the result of variables that are unable to be controlled. However, the results of this study are corroborated by the conclusions of previous works [16,32] that show that willingness to undergo vaccinations (including COVID-19 vaccination) is correlated to trust in research, in vaccines, and in healthcare professionals among the general population. It is also possible that we may have to take into account the idea that, in the case of vaccination, what is considered relevant is not the clinical act of administering the vaccine but the nature, the effectiveness, and safety of the vaccine itself. This means that it is the scientific research itself that may not be trusted by people and not the healthcare workers, who are considered to be the most reliable source of vaccine-related information for patients [33]. Indeed, according to Dubè et al., one of the main predictors of vaccine acceptance is the recommendation of the vaccination by a health care professional [34]. These potentially different attitudes between trust in science and trust in clinical practice may explain one of the dimensions of vaccine hesitancy.

Lastly, the data used here were collected before the start of the COVID-19 pandemic, so the results should be interpreted in that context. If available by 2020, it will be interesting to analyze future results, and further studies could be also conducted to compare our results with the current level of vaccine hesitancy related to pandemic context.

On the other hand, one of the main strengths of this study is the integration of descriptive and inferential statistics to describe the socio-cultural profiles associated with different perspectives concerning the harmfulness of vaccines. The use of a logistic regression model made it possible to analyze the potential confounders and to investigate the marginal effects of trust in the scientific community in the two ESS editions. Another strength was the large sample size interviewed for the two ESS surveys together with the inclusion of respondents from the general population. Finally, since the data were obtained from a CAPI survey (i.e., face-to-face with a pc support), the answers and results are more solid and reliable. In fact, it has been proven that face-to-face interviews are more successful in populations where a lack of awareness or understanding of vaccination is identified as a barrier [35].

## 5. Conclusions

Our work showed that in Italy from 2017 to 2019, the trust in the scientific community and, in particular, positive attitudes towards vaccinations grew significantly, as if to indicate a paradigm shift in public opinion compared to the past. Eventually, all of the information initiatives and communication campaigns conducted by the scientific community and research bodies alongside with coercive interventions implemented by the Italian Government might have sharply contributed to such an increase. Further central actions and effective public communication strategies might lead to a decrease in the phenomena of vaccine hesitancy and refusal and could be essential to garner public opinion and trust towards the scientific community.

## Figures and Tables

**Figure 1 vaccines-09-01206-f001:**
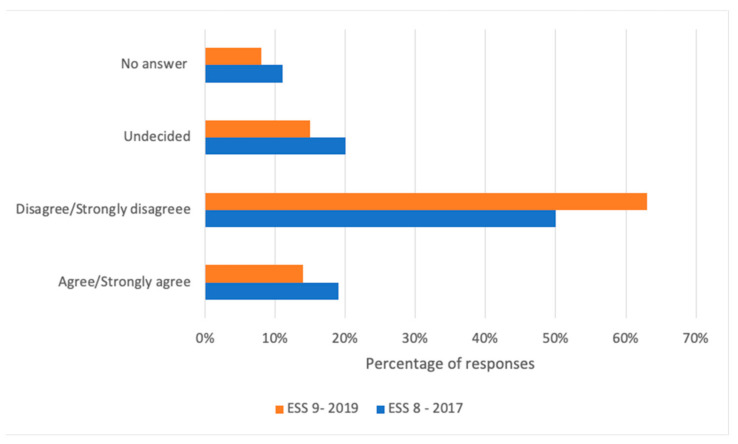
Comparison of the opinion concerning the statement ‘‘Vaccines are harmful and expose people to various diseases” in the European Social Surveys round 8 (2017) and round 9 (2019).

**Figure 2 vaccines-09-01206-f002:**
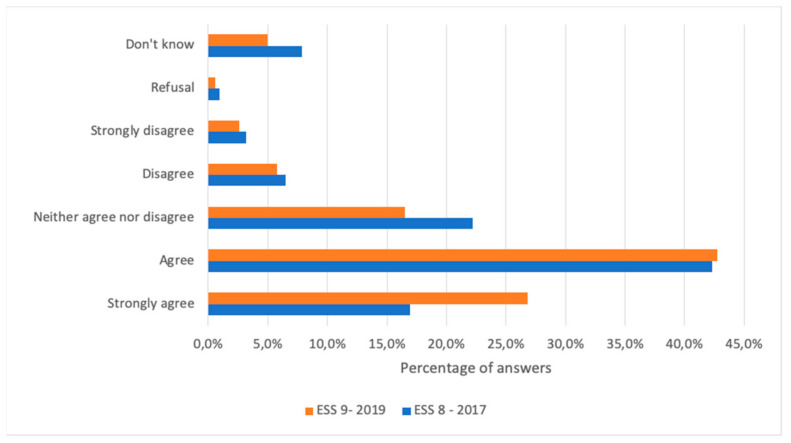
Comparison of the opinion concerning the statement ‘‘‘With regard to vaccines, recommendations from the scientific community can be trusted” in the European Social Surveys round 8 (2017) and round 9 (2019).

**Figure 3 vaccines-09-01206-f003:**
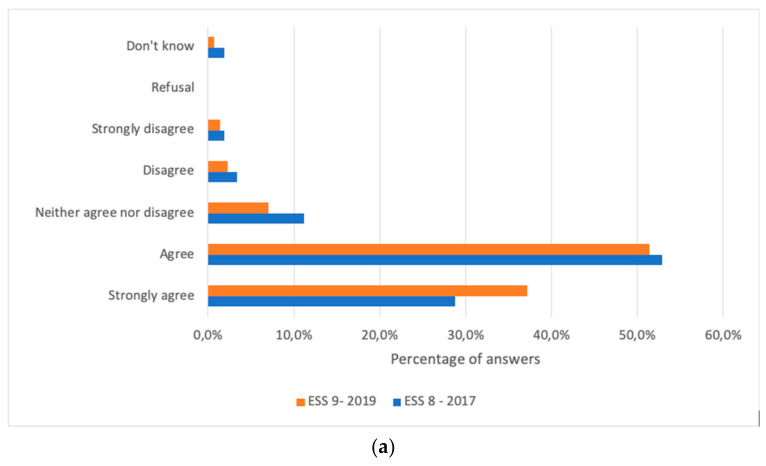
Comparison opinions concerning the statement ‘‘With regard to vaccines, recommendations from the scientific community can be trusted” (**a**) among those who consider vaccines to be safe and (**b**) among those who consider vaccines to be harmful in European Social Surveys Round 8 (2017) and Round 9 (2019).

**Figure 4 vaccines-09-01206-f004:**
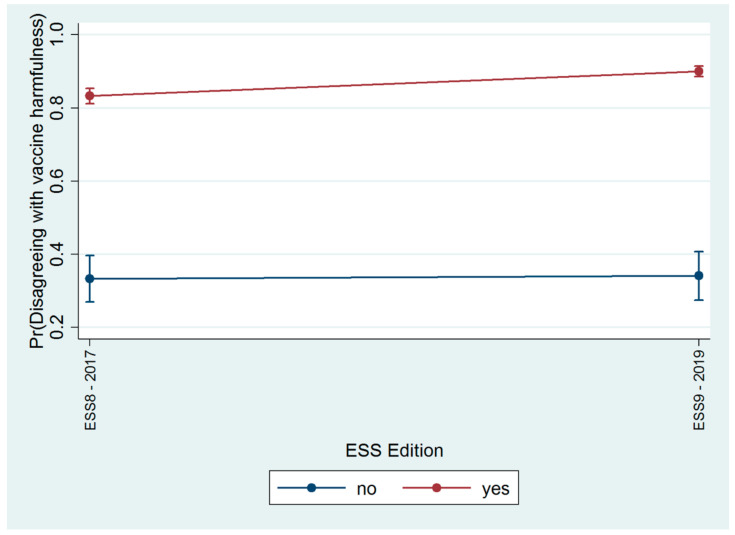
Marginal probability of disagreeing about the harmfulness of vaccines among those who do not trust the scientific community and those who do trust the scientific community in the two editions of the ESS survey (ESS8-2017 and ESS9-2019).

**Table 1 vaccines-09-01206-t001:** Probability of disagreeing with the harmfulness of vaccines associated with trust in the scientific community, age, gender, edition, and ESS edition. Probability expressed in terms of odds ratio with 95% confidence interval (95% CI).

	OR (95% CI)
** *Gender* **	
Males (reference)	1.00
Females	1.22 (0.93–1.37)
** *Age* **	
18–24 (reference)	1.00
25–44	0.74 (0.52–1.06)
45–64	0.77 (0.55–1.07)
65+	1.07 (0.75–1.53)
** *Education* **	
Low (reference)	1.00
Medium	1.57 (1.26–1.95)
High	2.41 (1.75–3.31)
** *Trust in scientific community* **	
No (reference)	1.00
Yes	10.47 (7.55–14.52)
** *Ess edition* **	
2017 (riferimento)	1.00
2019	1.04 (0.68–1.58)
** *Interaction term* **	
trust in scient. comm. (yes) × Edition (2019)	1.76 (1.10–2.83)

## Data Availability

The data that support the findings of this study are available upon request from the corresponding author, M.S.

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
