# Peer review of "Vaccine Hesitancy and Trust in the Scientific Community in Italy: Comparative Analysis from Two Recent Surveys"

_vaccines, 2021, doi:10.3390/vaccines9101206_

Round 1
Reviewer 1 Report
Vaccine hesitancy – and factors influencing it – represents a hot topic in vaccination. Furthermore, vaccine hesitancy – already included among the 10 major threats for global health- has become increasingly relevant in the fight against the SARS CoV-2 pandemic. Accordingly, the paper of Cadeddu and co-workers (that reports on the changes in vaccine hesitancy between two surveys performed by the European Social Survey two-year apart, and variables influencing these changes including trust in the scientific community), represents a valuable information for Vaccine readers.
However, as correctly mentioned by the authors among weaknesses of the paper, the expectation aroused by the title is partly disappointed by the fact that the two surveys were performed before the COVID-19 vaccination campaign, in a selected (Italian) population, and refer to vaccination in general and not to a specific vaccine.
Despite these limitations, the paper is well written, the methodology is accurate and the references adequate.
Tables 1a and 1b, reporting on the descriptive characteristics of respondents in relation to harmfulness of vaccines in the two survey, may be redundant for the message of the paper. The valuable information included in the two tables might be part of the supporting online information and replaced by graphs focussing on the most significant changes observed for some of the characteristics analysed.
Author Response
Response to Reviewer 1 Comments
Point 1: Vaccine hesitancy – and factors influencing it – represents a hot topic in vaccination. Furthermore, vaccine hesitancy – already included among the 10 major threats for global health- has become increasingly relevant in the fight against the SARS CoV-2 pandemic. Accordingly, the paper of Cadeddu and co-workers (that reports on the changes in vaccine hesitancy between two surveys performed by the European Social Survey two-year apart, and variables influencing these changes including trust in the scientific community), represents a valuable information for Vaccine readers.
However, as correctly mentioned by the authors among weaknesses of the paper, the expectation aroused by the title is partly disappointed by the fact that the two surveys were performed before the COVID-19 vaccination campaign, in a selected (Italian) population, and refer to vaccination in general and not to a specific vaccine.
Despite these limitations, the paper is well written, the methodology is accurate and the references adequate.
Tables 1a and 1b, reporting on the descriptive characteristics of respondents in relation to harmfulness of vaccines in the two survey, may be redundant for the message of the paper. The valuable information included in the two tables might be part of the supporting online information and replaced by graphs focussing on the most significant changes observed for some of the characteristics analysed.
Response 1: Thanks for your valuable suggestions. We added Table1a (Table S1) and Table1b (Table S2) as Supplementary material and, in the results section, we replaced them with four graphs which address the two central issues: trust in vaccines and in the scientific community.

Reviewer 2 Report
The manuscript "Vaccine hesitancy and trust in the scientific community in Italy: comparative analysisfrom two recent surveya" is a well-written paper investigating the vaccination attitude of Italians as reported in the European Social Surveys conducted in 2017 and in 2019.
The main thesis of the paper is that scientific trust is the main factor influencing vaccine hesitancy. The only relevant negative aspect of the paper is that when you read it you have the feeling of a study which takes a picture of a pre-Covid situation regarding vaccine hesitancy that might be different from current attitudes towards vaccination.
The methods are sound and the argumentation proposed by the authors is clear. I think that the paper can be published. Nonetheless, I list some minor remarks:
- Please, state already in the Introduction that the one of the aims of the paper is to investigate the attitudes of Italian people towards vaccination in a pre-Covid context. This is important since in this way it will be possible to understand what it is happaning nowadays. The main aim is of course a comparison between EES8 and EES9 regarding vaccine hesitancy shifts, but the possibility in the future to compare these results with current vaccine hesitancy adds values to the paper.
- There exists extensive literature showing that clinicians are the professionals with the higest level of credibilty among professionals. However, in the case of vaccination what is considered relevant is not the clinical act of administering the vaccine, but the nature of vaccine itself. This means that it is the scientific research that may be not trusted by people. These potential different attitudes between trust in science and trust in clinical practice may explain one o fthe dimension of vaccine hesitancy.
- p. 13. line 248. Emotional aspects of misinformation are mentioned in correlation to vaccine hesitancy. This is an interesting point that may be extended a bit. In the sense that emotions as well values and the proper frame of medical information can provide the proper ground for individual decisions toward vaccination. For an analysis of values in health care, see, for instance: Daniele Chiffi (2021). Clinical Reasoning: Knowledge, Uncertainty, and Values in Health Care. Cham: Springer.
- The are some typos when starting a new line at lines: p. 10, lines160/161; p. 12, lines198/199;
- p. 15, line 340. I would not say "in most recent years". It is better to indicate the specific years of the investigation, in order to avoid the possibility that reader may associate these finsings to the ones of the vaccination for Covid-19.
Author Response
Response to Reviewer 2 Comments
Point 1: Please, state already in the Introduction that one of the aims of the paper is to investigate the attitudes of Italian people towards vaccination in a pre-Covid context. This is important since in this way it will be possible to understand what it is happening nowadays. The main aim is of course a comparison between EES8 and EES9 regarding vaccine hesitancy shifts, but the possibility in the future to compare these results with current vaccine hesitancy adds values to the paper.
Response 1: Thanks for your valuable suggestion. We added your indications in the introduction in order to make the objectives more clear and we specified that further studies could be conducted in order to compare our results with the current vaccine hesitancy related to pandemic context. Please see lines 95;99-100; 384-385.
Point 2: There exists extensive literature showing that clinicians are the professionals with the highest level of credibility among professionals. However, in the case of vaccination what is considered relevant is not the clinical act of administering the vaccine, but the nature of vaccine itself. This means that it is the scientific research that may be not trusted by people. These potential different attitudes between trust in science and trust in clinical practice may explain one of the dimensions of vaccine hesitancy.
Response 2: We thank the reviewer for the comment. We clarified in the discussion this important aspect of vaccine hesitancy which is necessary to take into consideration. Please see line 372-380.
Point 3: p. 13. line 248. Emotional aspects of misinformation are mentioned in correlation to vaccine hesitancy. This is an interesting point that may be extended a bit. In the sense that emotions as well values and the proper frame of medical information can provide the proper ground for individual decisions toward vaccination. For an analysis of values in health care, see, for instance: Daniele Chiffi (2021). Clinical Reasoning: Knowledge, Uncertainty, and Values in Health Care. Cham: Springer.
Response 3: Thank you for your interesting suggestion. The concept of how emotions can provide the proper ground for individual decisions toward vaccination was deepened in the discussions, thanks to the thought-provoking book you suggested to analyze. Please see line 278-295.
Point 4: The are some typos when starting a new line at lines: p. 10, lines160/161; p. 12, lines198/199;
Response 4: Thank you for your warning, we agreed with you and modified the text.
Point 5: p. 15, line 340. I would not say "in most recent years". It is better to indicate the specific years of the investigation, in order to avoid the possibility that reader may associate these findings to the ones of the vaccination for Covid-19.
Response 5: Thanks for pointing it out, it has been amended in the text. Please see line 399.